# Selection for constrained peptides that bind to a single target protein

Andrew M. King[1,2], Daniel A. Anderson [1,3], Emerson Glassey[1,3], Thomas H. Segall-Shapiro[1], Zhengan Zhang[1], David L. Niquille[1], Amanda C. Embree[2], Katelin Pratt[2], Thomas L. Williams [2], D. Benjamin Gordon [1,2] & Christopher A. Voigt [1,2✉]

Peptide secondary metabolites are common in nature and have diverse pharmacologically-relevant functions, from antibiotics to cross-kingdom signaling. Here, we present a method to design large libraries of modified peptides in *Escherichia coli* and screen them in vivo to identify those that bind to a single target-of-interest. Constrained peptide scaffolds were produced using modified enzymes gleaned from microbial RiPP (ribosomally synthesized and post-translationally modified peptide) pathways and diversified to build large libraries. The binding of a RiPP to a protein target leads to the intein-catalyzed release of an RNA polymerase σ factor, which drives the expression of selectable markers. As a proof-of-concept, a selection was performed for binding to the SARS-CoV-2 Spike receptor binding domain. A 1625 Da constrained peptide (AMK-1057) was found that binds with similar affinity (990 ± 5 nM) as an ACE2-derived peptide. This demonstrates a generalizable method to identify constrained peptides that adhere to a single protein target, as a step towards "molecular glues" for therapeutics and diagnostics.

[1] Synthetic Biology Center, Department of Biological Engineering, Massachusetts Institute of Technology, Cambridge, MA, USA. [2] Broad Institute of MIT and Harvard, Cambridge, MA, USA. [3]These authors contributed equally: Daniel A. Anderson, Emerson Glassey. ✉email: cavoigt@gmail.com

Short peptides have vast potential as therapeutics due to their intrinsic ability to bind to protein targets and the simplicity of creating combinatorial libraries. However, linear peptides are generally not good drugs due to their conformational flexibility, inability to cross the cell membrane, and sensitivity to proteases[1]. Bacteria overcome these issues by chemically modifying peptides to introduce functional groups and constrain their structures to improve affinity and stability. The modified peptides are secreted and act on eukaryotic cells by binding to surface proteins or crossing their membrane to interact with intracellular targets[2–4]. The peptides can inhibit cellular processes by occluding the active site of an enzyme or blocking a protein–protein interaction by binding to the interface. They can also act as a "molecular glue" to bring two proteins together, the most well-known example being rapamycin, which binds to two human proteins involved in signaling in order to suppress the host immune response[5]. Secreted peptides evolve to bind new targets as bacteria adapt to environmental niches[6–8].

Libraries of linear peptides can be easily created using combinatorial chemistry or by encoding them as genes using oligonucleotide pools and expressing them in vivo[9]. Sometimes, it is possible to add modifications to a linear peptide that is a natural substrate or emerges from a screen, but this can disrupt affinity[10]. Cells produce modified peptides through large non-ribosomal peptide synthetases (NRPSs) or by expressing them as genes that are post-translationally modified (RiPPs)[11,12]. Many pharmaceuticals have been derived from NRPS products, but the synthetases are notoriously difficult to genetically diversify[13–15]. Therefore, libraries of NRPS analogs have been built using synthetic chemistry, which can be used to diversify the amino acids around a scaffold constrained by cycles and other modifications[16]. For example, rapamycin is produced by an NRPS and a library of 45,000 macrocycles was chemically synthesized to expand the targets beyond mTOR[17]. In contrast, RiPPs are encoded as genes, thus simplifying the creation of libraries of diverse peptides[18]. Once a precursor peptide is expressed, modifying enzymes bind to a leader or follower sequence and modify the core, which is then proteolytically released[11]. Libraries of RiPPs have been generated in heterologous hosts, including E. coli, and can be produced using cell-free protein synthesis (CFPS)[19,20].

Screening methods have been developed to find functional peptides within large libraries. If each peptide variant is purified to high concentration, they can be individually screened using cell-based or biochemical assays[21]. Depending on the assay complexity, up to tens of thousands of variants can be evaluated through automation[22]. Alternatively, the peptides can be pooled and "panned" for those that bind to a protein target. By fusing the peptide to its genetic material (e.g., with mRNA-peptide fusions or phage display), the sequences of binders can be determined. It is more difficult to combine peptide modifications with these techniques, but innovative approaches have been applied to cyclize displayed peptides[23–31].

Screens have been developed to identify compounds that disrupt protein–protein interactions by binding to their interface[32]. One approach is to fuse each protein to one half of a split reporter (luciferase, GFP, or T7 RNAP), but the disruption of this interaction leads to the loss of signal which is less desirable for screens[33–37]. This is addressed by reverse two-hybrid systems, where the proteins are fused to the DNA-binding domains of repressors, such that the disruption of their interaction leads to the derepression of a promoter[38]. This can be used to drive a selection that links the disruption of the interaction with the survival of the cell, thus allowing orders of magnitude more variants to be evaluated in an experiment. This has been applied to finding cyclized peptides and RiPPs that are μM inhibitors of the Gag-TSG1010 and p6-UEV interactions, respectively, that are critical for the human immunodeficiency virus (HIV) life cycle[39,40].

For many applications, it is desirable to identify a peptide that binds to a single target. For example, this could be to detect a protein as a diagnostic[41,42] or to degrade a therapeutic target by directing it to cellular proteolytic machinery[43,44]. Here, we present a selection strategy to identify RiPPs that bind to a single target-of-interest, without specifying its binding position. A synthetic genetic circuit was constructed in E. coli that responds when a modified peptide binds to a bait protein. A RiPP, including leader and core sequence, was fused to one half of a split intein and split σ factor and the target protein was fused to the other half of the intein and σ factor. When the RiPP binds the target, the intein covalently releases the complete σ factor, which turns on a promoter to drive a selectable marker. This approach was demonstrated by identifying a synthetic RiPP constrained by a thioether macrocycle that binds to the Spike receptor binding domain (RBD) from the severe acute respiratory syndrome coronavirus-2 (SARS-CoV-2)[45]. Rounds of selection were performed to identify a hit, from which a 14 amino acid modified sequence is derived that binds to a unique site that is outside of the regions bound by the native human substrate (ACE2) and neutralizing antibodies ($K_D = 990 \pm 5$ nM). This demonstrates the design of cells to simultaneously make peptidic small molecules and evaluate their ability to bind to a defined therapeutic target, thus allowing billions of compounds to be tested in parallel in single experiments.

## Results

**Genetic circuit to detect peptide binding to a target protein.** The circuit converts a binding event into a transcriptional response. It is based on two fusion proteins containing the RiPP and the bait that, upon binding, bring together two halves of a split intein that catalyze the release of a σ factor that recruits RNA polymerase (RNAP) to a promoter (Fig. 1a). We selected a version of the split intein from Nostoc punctiforme PCC73102 (Npu) that has been artificially fused and re-split at a different position to produce large N-terminal and short C-terminal fragments. This version has much less activity compared to the native intein's exceptional stability and splicing kinetics[46,47]. The reduced association between fragments facilitates their use as a protein–protein interaction detector. The σ factor ECF20_992 was chosen because of its strong transcriptional output and sensitivity to linker variations between the DNA binding domains[48,49]. The first fusion protein consists of the N-terminal σ factor fragment ($\sigma^N$) placed upstream of the N-terminal Npu fragment ($Npu^N$) and linker followed by the bait protein. The second fusion protein has the C-terminal Npu fragment ($Npu^C$), followed by the C-terminal σ factor fragment ($\sigma^C$). The N-terminus of this protein has the leader peptide that recruits modifying enzymes and the core sequence of the RiPP followed by a flexible 20 aa linker (Methods). Successful splicing of full length σ factor leads to the activation of the $P_{20\_992}$ promoter, thereby turning on the downstream genes.

To develop the circuit, we selected the well-studied interaction between the proteins p53 and Mdm2 as a test case[50]. Specifically, we used residues 17-124 of Mdm2 (Mdm2*) and a high affinity ($K_D = 0.5$ nM) variant of residues 15–29 of p53 (PMI)[51,52] as bait and peptide fusions to the $\sigma^N$-$Npu^N$ and $Npu^C$-$\sigma^C$ fragments, respectively. The two genes were placed under the control of 3OC6-AHL-inducible $P_{LuxB}$ and IPTG-inducible $P_{Tac}$ promoters in E. coli Marionette Clo[53]. For characterization, gfp was cloned downstream of $P_{20\_992}$. The expression of the two fusion proteins was controlled using 3OC6-AHL and IPTG and the

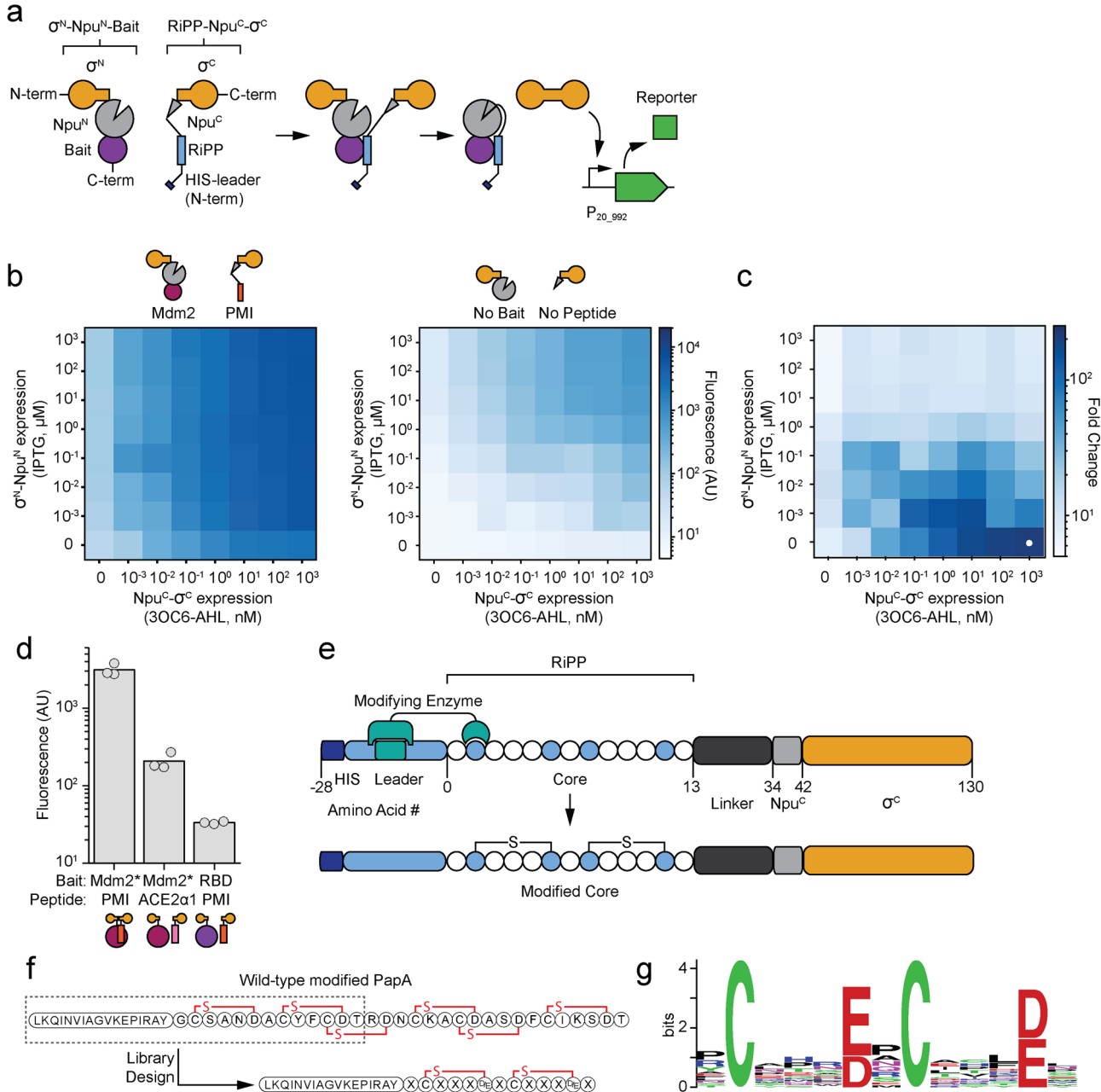

**Fig. 1 Split intein system for in vivo detection of protein–protein interactions. a** A schematic is shown for the detection of a binding event between a RiPP and target. A split intein from *Nostoc punctiforme* PCC73102 (Npu), is used to splice together a σ factor for downstream transcription. **b** Inducible interaction-mediated splicing. The median fluorescence is shown as a function of the expression of the two halves of the sensor proteins. The induction of PMI- and Mdm2*-driven association (left) or split intein alone (right) are shown. The plots are a representative replicate selected from three performed on different days. **c** The specificity is calculated using the data in part b: (Fluorescence$^{Mdm2^*+PMI}$/Fluorescence$^{no\ bait+no\ peptide}$). The white dot marks the highest fold-change in expression. **d** The fluorescence measured from the circuit containing a binding pair (Mdm2*:PMI) and non-binding pairs are compared. The 3OC6-AHL concentration for all inductions was 1 μM. Three replicates performed on different days are shown. **e** A schematic of the RiPP-containing half of the split intein system is shown, with the modified core residues in blue. **f** The structure of the wild-type PapA modified peptide[56] is shown with a box around the region used to design the peptide library and the resulting scaffold sequence. **g** Library sequence weblogo for the 14 modified library variants observed (Methods).

fluorescence output of the circuit was measured using flow cytometry (Methods). When the interaction was driven by Mdm2* and PMI, there was a large induction over a wide range of expression levels (Fig. 1b). However, when there was no driving protein–protein interaction, the split inteins and split σ factors only turned on the output promoters at high levels of expression. When performing a selection, it is important that the background expression of the σ$^N$-Npu$^N$ and Npu$^C$-σ$^C$ fragments

not induce the circuit. The difference between the presence and absence of a peptide:bait interaction was maximized to >200-fold when Npu$^C$-σ$^C$ was fully induced and σ$^N$-Npu$^N$ fragment uninduced (Fig. 1c). To remove the need to add inducer, σ$^N$-Npu$^N$-bait was placed under the control of a weak constitutive promoter. The peptide-Npu$^C$-σ$^C$ was kept under 3OC6-AHL control so that its concentration can be decreased for progressive rounds of selection to bias towards higher-affinity binders.

The inducible range of the sensor was then determined when either the bait or peptide were changed to disrupt the interaction (Fig. 1d). When a peptide based on the N-terminal residues 19-56 of ACE2 (ACE2α1)[54] was used, which does not bind to the Mdm2* target, the fluorescence output of the circuit drops 15-fold. Similarly, when the target peptide was swapped to be residues 328-533 of the SARS-CoV-2 Spike protein (RBD)[45], to which PMI does not bind, the fluorescence output of the circuit was almost two orders-of-magnitude lower. Using published affinity values for PMI variants[52], we estimated that the threshold for detection of this system is 5 μM, with a linear increase in fluorescence to 10 nM. Outside of this range, the circuit cannot distinguish variants (Supplementary Figure 1).

The peptide needs to be able to be modified by RiPP enzymes in the context of its fusion to C-terminal Npu$^C$-σ$^C$ (Fig. 1e). RiPP modifying enzymes bind to an N-terminal leader sequence and then modify the core sequence. This process is not typically influenced by the addition of C-terminal fusions, including entire proteins[24,26,55]. RiPP cores are released from the leader by proteases[11], which were not included in our system. To improve the accessibility of the core, we added a 20 amino acid linker (GGKGGPGGRGGVGGGGGIGG) between it and the split intein.

We performed a preliminary experiment to test whether a RiPP enzyme could modify a large fraction of core sequences in a library in the context of the fusion protein. The PapA/PapB peptide/enzyme pair from the *Paenibacillus polymyxa* freyrasin biosynthetic gene cluster was selected[56]. PapB introduces a thioether macrocycle between core C and D/E residues and it was shown to be tolerant to amino acid diversity at the unmodified residues[56,57]. Based on this system, we designed a simplified core and leader peptide containing two cycles (Fig. 1f). A library was constructed allowing full (NNK) degeneracy at 9 unconserved amino acid positions and D/E at the two macrocyclized positions, resulting in $10^{12}$ theoretical peptides. We selected 19 random members from this library, expressed them with PapB, and evaluated cyclization by measuring the mass shift observed using liquid chromatography-mass spectrometry (LC-MS) (Methods). From this subset, 16 complete peptides were observed and 14 had masses consistent with either one or two macrocycles (Supplementary Table 4). The modification did not bias the core positions towards specific amino acids (Fig. 1g).

**Complete selection system.** The genetic system used for the selections, involving seven genes, is shown in Fig. 2a. It has been previously demonstrated that the SARS-CoV-2 Spike RBD can be expressed in *E. coli*[58,59]. This domain was used as part of σ$^N$-Npu$^N$-RBD, which is placed under the control of the weak BBa_J23105 constitutive promoter. The DNA encoding the peptide library (*ripp*) was used to build a fusion library (*ripp-npu$^C$-σ$^C$*), the expression of which was controlled with 3OC6-AHL. The modifying enzyme was placed under the control of the cumate-inducible promoter. When the σ factor is released, the P$_{20\_992}$ promoter drives the expression of an operon containing the selectable marker fusion *gfp-cat*. This enabled positive selection through the addition of chloramphenicol (Cm) to the media. The operon also encodes *pheS* because it can be used in negative selections[60], but it was not used in this study.

**Library design and selection.** The selection scheme is shown in Fig. 2b. Each cell makes a different variant of the peptide. When it binds to the target, *gfp-cat* is expressed. Over multiple rounds of selection, increased stringency was applied by increasing the concentration of Cm or decreasing peptide induction (by decreasing the concentration of 3OC6-AHL) (Fig. 2c).

The library was based on a simplified PapB-modified 13 amino acid core structure (Fig. 2d). The positions not modified by PapB were diversified using NNK codons, leading to a theoretical library size of $10^{12}$ variants. When the library was introduced to cells using electrocompetence, it was limited in practice to $10^8$ per transformation. Rounds of positive selection were performed; after each round, the plasmids from the surviving cells were isolated and retransformed (Fig. 2c) (Methods). Cells were grown overnight in increasing concentrations of Cm: Round 1 (300 μM), Round 2 (800 μM), and Round 3 (1200 μM). GFP expression was measured after each round using flow cytometry, showing a continuous increase in the fluorescence output (Fig. 2e).

When using 300 μM Cm for the initial selection, two peaks were observed: one lower (2,000 AU) and one higher (20,000 AU). This higher peak was attributed to escape mutants from the selection plasmid breaking. To eliminate escapes from round to round, we digested the selection plasmid and re-transformed enriched peptide plasmids into the expression strain. This eliminated the peak corresponding to escapes (Fig. 2e). All selection rounds were then analyzed using next-generation sequencing (NGS). The number of unique RiPP sequences decreased after each round, indicating enrichment: 139,320, 88,229, and 63,344. The abundance of each sequence was calculated and 32 were found to represent >1% of the population after Round 3. These were further reduced to 20 by only considering those that showed consistent enrichment between rounds 1→2 and 2→3.

Next-generation sequencing (NGS) was used to analyze the product of each selection, rather than physically isolating the top variants (Methods). Because of this, we had to re-obtain the top 20 hits using DNA synthesis. Each was cloned into the *ripp-npu$^C$-σ$^C$* plasmid and re-assessed for activity. This process eliminates any cheaters that may have arisen throughout the selection process due to mutations to the plasmid or genome. These constructs were transformed into strains containing the modifying enzyme and either Spike RBD or Mdm2* as bait, with the latter selected to measure off-target binding. The fluorescence output of the circuit was measured using flow cytometry under the same growth conditions and inducer concentrations used for the selections. From these, the core VCKYGEWCEIVEI encoded within Pap2c_1 demonstrated a strong fluorescence output and 14-fold specificity for the Spike RBD as compared to Mdm2* (Fig. 2f).

Variants of Pap2c_1 were constructed and the impact on binding to Spike RBD evaluated using the fluorescent reporter and flow cytometry (Methods). Any truncations from the N-terminus of the core sequence led to lower fluorescence (Fig. 3a, Supplementary Fig. 2). However, even as few as six amino acids still showed a comparable binding profile (up to 3-fold higher fluorescence output of the circuit) than the off-target control PMI (dashed lines in Fig. 3a and data in Fig. 1d). These truncations lead to an affinity equivalent to ACE2α1 binding to Spike RBD, which has been reported to bind with low μM affinity[54].

Next, single-site saturation mutagenesis (NNK) was conducted for each residue of the core. The 15 libraries, each corresponding to a different residue, were pooled and enriched for binding using the selection (250 μM Cm) while inducing Pap2c_1 (3.2 nM OC6-AHL). The selected and unselected libraries were analyzed by NGS and the enrichment of each amino acid substitution calculated (Methods) (Fig. 3b). The majority of single amino acid substitutions led to lower affinity for RBD, indicating that a near-optimal sequence was identified in the selection (Fig. 3b). Substitutions at residue 3 from alanine to cysteine or branched hydrophobic amino acids (I, L, V) led to greater enrichment. These also led to an increase in fluorescence when measured by cytometry (Fig. 3c).

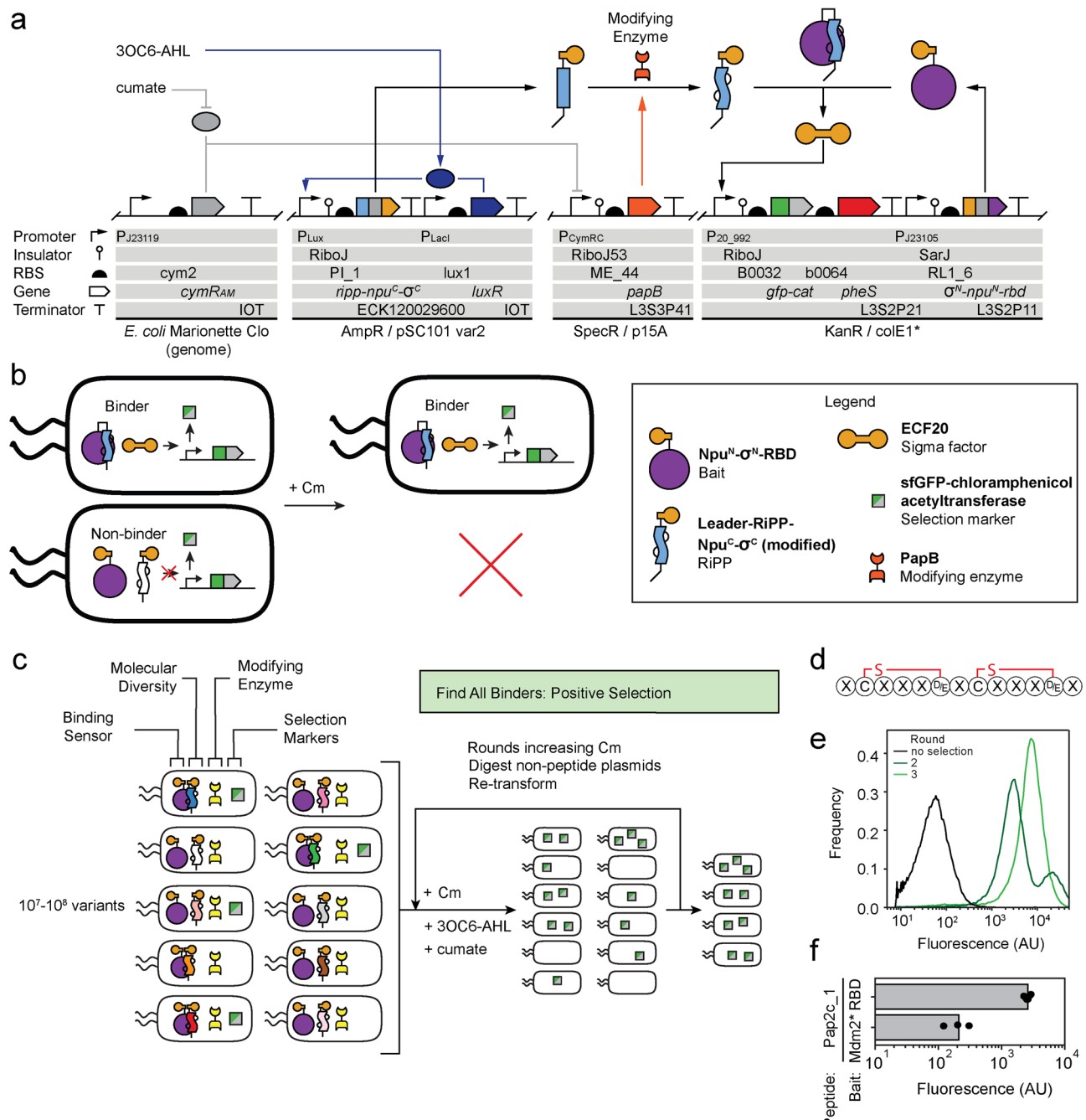

**Fig. 2 Selection system to find RBD-binding RiPPs. a** The genetic circuit diagram for the RBD-binding RiPP selection system is distributed across three plasmids and two genomic regions. Two small molecules: 3OC6-AHL and cumate are used to control the expression of the RiPP peptide and modifying enzyme, respectively. **b** Schematic of selection output under two conditions: with an RBD-binding RiPP and without an RBD-binding RiPP. Binding is shown to result in the production of GFP-chloramphenicol acetyltransferase (GFP-CAT; green/gray squares). **c** Overview of the positive selection. Selection rounds were conducted in the presence of the RiPP peptide and modifying enzymes and used increasing Cm concentrations for increased stringency. **d** The core scaffold for the Pap2c library (IbAMK-103). Predicted macrocles are colored between constrained residues and "X" residues correspond to NNK translated amino acids. **e** Cytometry distributions for positive selections on the Pap2c library beginning with no selection, round 2, and round 3 of positive selections (0, 800, and 1200 μM Cm). Fluorescence of the GFP fused to CAT is reported. **f** RBD-specific hit (Pap2c_1) isolated from genetic selection. Pap2c_1 was used as peptide against the non-specific bait (Mdm2*) and specific bait (RBD). 3OC6-AHL (1 μM) was used for inductions and three replicates performed on different days are shown.

Next, we tested how the insertion of a TEV protease site (ENLYFQG) between the core and leader impacts affinity (Fig. 3d, Supplementary Figure 2). This leads to 10-fold lower fluorescence that can be recovered by moving the cleavage site four amino acids (IRAY) upstream of the core. The lower affinity may be due to the disruption of an α-helix (Supplementary Figure 3).

**AMK-1057 binding to Spike RBD**. To aid with downstream purification of peptides that emerge from the selection, the leader and core were fused to the N-terminal RiPP stabilization tag, RST$_N$. TEV protease cleavage sites (ENLYFQG) are placed between RST$_N$ and the leader and between the leader and the core (Fig. 4a). The resulting fusion protein is referred to as

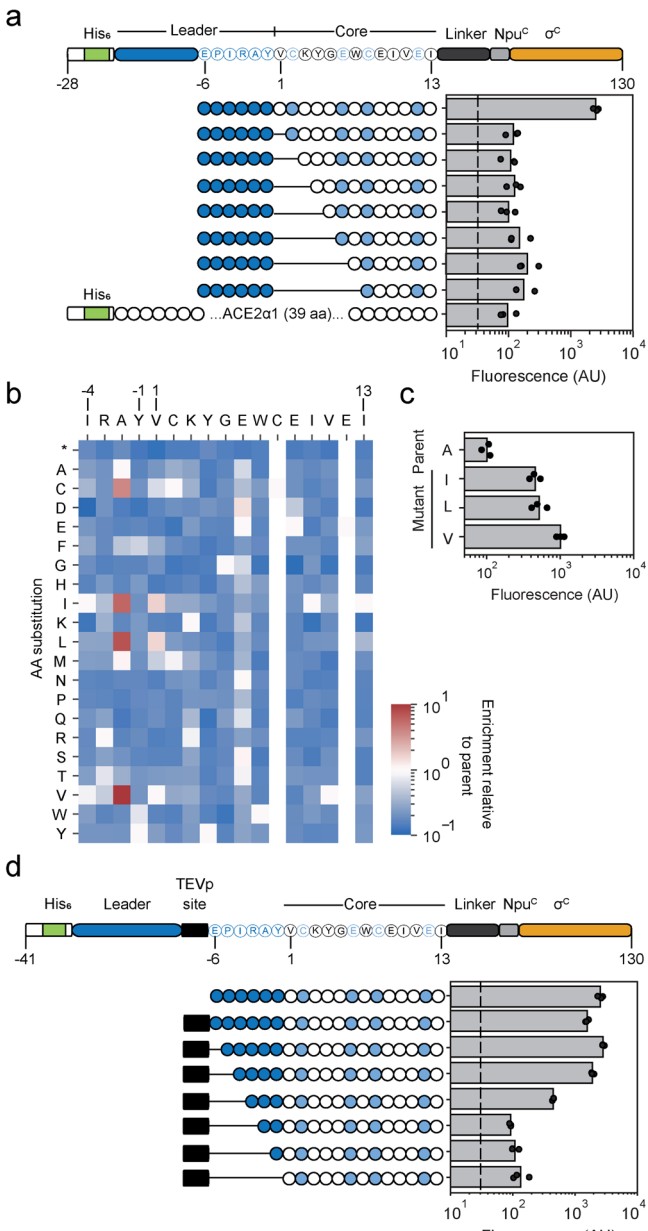

**Fig. 3 Sequence-activity profiling of Pap2c_1 yields better binders in vivo.** **a** Schematic of Pap2c_1 peptide fusion used for in vivo measurements. Leader and core amino acids considered for profiling are displayed and predicted modified core residues highlighted in light blue. A fusion protein of the 39 amino acid ACE2α1 peptide to the C-terminal NpuC-σC was included as an off-target control. Dashed lines indicate the fluorescence output of the circuit driven by the RBD:PMI interaction measured in Fig. 1d. 3OC6-AHL (1 μM) was used for inductions and three replicates performed on different days are shown. **b** Heat map of variant enrichment from selection and NGS. Amino acid substitutions enriched relative to the parent sequence are colored red. **c** Fluorescence output of consensus variant strains constructed based on data from (b). Individually constructed variants at position -3 (A, I, L, V) were evaluated for fluorescence output of the circuit. 3OC6-AHL (0 μM) was used for inductions and three replicates performed on different days are shown. **d** Schematic of the Pap2c_1 peptide fusion used for in vivo measurements and TEV protease site insertion analysis. Modified core residues are shown in light blue. Dashed lines indicate the fluorescence output of the circuit driven by the RBD:PMI interaction measured in Fig. 1d. 3OC6-AHL (1 μM) was used for inductions and three replicates performed on different days are shown.

RST$_N$_Pap2c_1. Note that when TEV protease is used to release the core, it leaves an additional glycine on its N-terminus, which did not emerge from the selection.

We expressed RST$_N$_Pap2c_1 and purified the modified core peptide in order to determine the location of its binding to cell-derived Spike RBD. The cleaved, modified peptide is referred to as AMK-1057. RST$_N$_Pap2c_1 was co-expressed at liter scale with the modifying enzyme, Ni-NTA purified, and cleaved using TEV protease (Methods). The cleaved peptide was isolated using solid phase extraction (SPE) and semiprep HPLC purification. Three peptides were isolated: leader (200 μg/L), unmodified core (640 μg/L) and AMK-1057 (360 μg/L) (Fig. 4b).

High-resolution LC-MS analysis of both modified (expected m/z: 1625.7338; observed m/z: 1625.7332) and unmodified (expected m/z: 1627.7494; observed m/z: 1627.7484) peptide showed a mass shift corresponding to formation of a single cycle, despite the library being based on a two-cycle scaffold (Fig. 4c). The macrocycle found in AMK-1057 is predicted to be formed through the covalent linkage of a side chain cysteine sulfur atom to the Cβ on the downstream glutamate residue, which is stable to standard collision-induced dissociation conditions[56]. This property was used to annotate the macrocycle placement using high-resolution tandem MS (HR-MS/MS) and hypothetical structure enumeration and evaluation (Fig. 4d)[61]. Fragmentation analysis indicated that the macrocycle forms at the C-terminal end of AMK-1057, between C9 and E13 (Fig. 4e).

We then performed in vitro binding experiments using Expi293F human cell-derived and purified Spike RBD. Bio-layer interferometry (BLI) was used to measure the affinity of AMK-1057 to Spike RBD as 990 ± 5 nM (Fig. 4f; Supplementary Fig. 2) (Methods). AMK-1057 is in the higher range of natural RiPPs binding to their target (e.g., lassomycin at 0.41 μM, microcin J25 at 2 μM) and some peptidic drugs (e.g., vancomycin at ~1 μM)[62–64]. Neither the purified unmodified core peptide nor the leader sequence bound to the target. We purified the ACE2α1 peptide and found that it bound to the Spike RBD with a $K_D$ of 647 ± 9 nM, which is similar to the reported value of 1.3 μM[54] (Fig. 4f, Supplementary Fig. 2).

To define a more specific region of the Spike RBD that AMK-1057 targets, BLI was used to measure binding in the presence of competitors with known epitope binding profiles[65]. Recombinant, human-expressed ACE2 (hACE2; $K_D$ = 44.2 nM)[66] and candidate therapeutic antibodies B38 ($K_D$ = 70.1 nM)[67] and CR3022 ($K_D$ = 6 nM)[68] bind to different regions of Spike RBD, visualized in Fig. 4g. BLI traces show association (300 seconds, pre-dashed line) of either hACE2, B38, or CR3022 with target RBD (Fig. 4h, i, j). After reaching steady-state, the biosensor containing bound complex was added to either control wells or AMK-1057 and an increase in Δnm observed as a measurement of target binding. In all three experiments, AMK-1057 associated with the RBD, indicating that hACE2, B38, and CR3022 do not occlude binding. Therefore, the binding epitope for AMK-1057 must be distinct from the regions highlighted in Fig. 4g (shown in gray).

## Discussion

This work introduces a technique to capture modified peptides that bind to a single target protein without specifying the binding location, which can be valuable for "undruggable" targets lacking an obvious binding pocket. This is in contrast to two-hybrid screens, where it is the disruption of a protein–protein interaction or a functional screen that requires blocking an enzymatic activity. Our approach could be useful in finding binders that disrupt a protein–protein interaction when one of the participating proteins cannot be expressed in functional form in a recombinant host or when binding to one of the two targets is

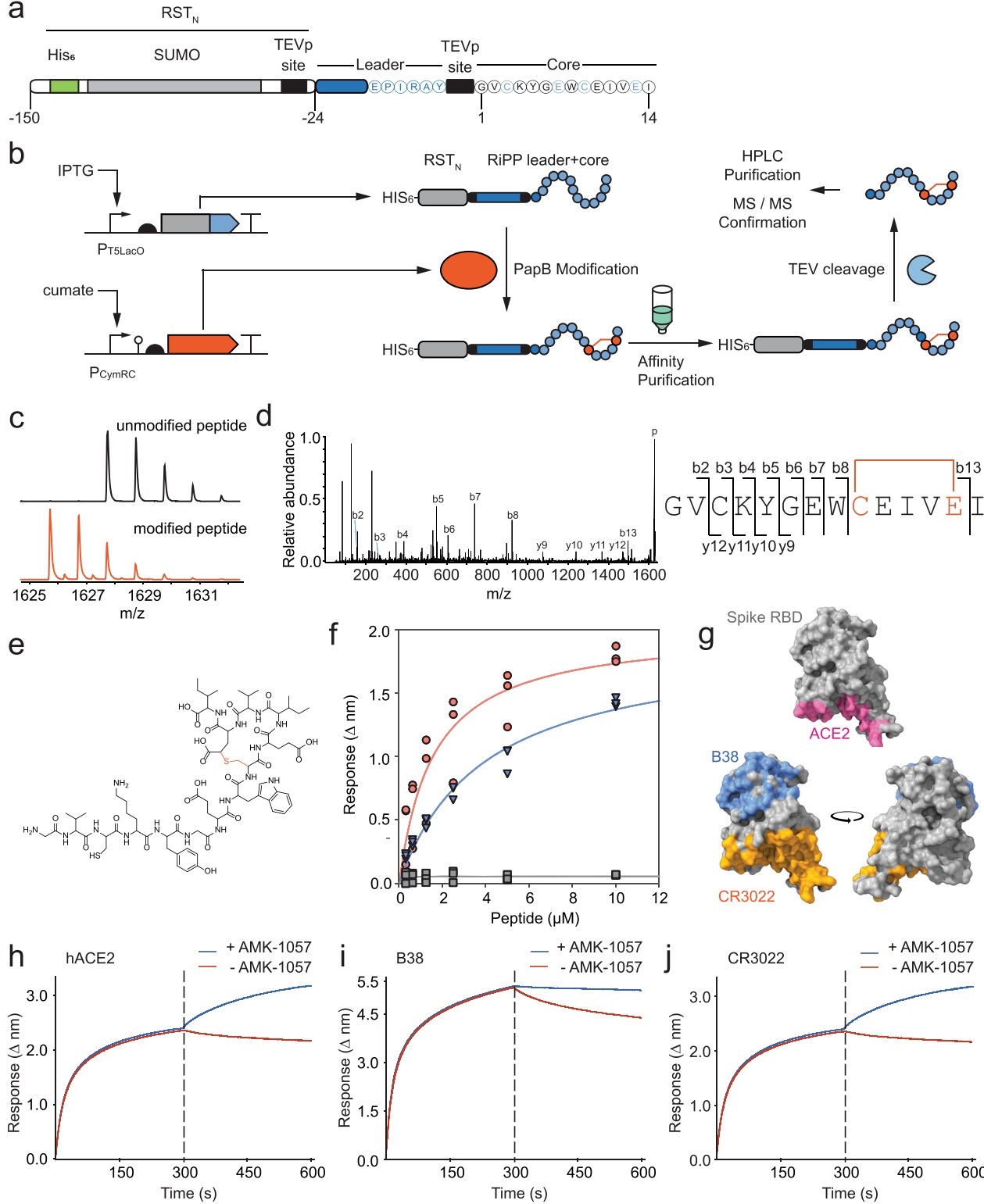

preferred. However, there is no guarantee that the peptide that emerges from the selection will bind to a therapeutic site at the interface. Indeed, while our original intent was to discover peptides that block Spike binding to ACE2, the peptide appears to bind in a region distinct from ACE2 and even outside of the regions bound to by therapeutic antibodies. Still, there are therapeutic modalities that do not require binding to a specific position, such as PROTACs (PROteolysis Targeting Chimeras)

that make use of a weak (as high as 10 μM) peptide binding-moiety to bring E3 ubiquitin ligase to the protein target[41–43,69–71]. The modified peptides could also be applicable to diagnostics, for example, when conjugated to gold nanoparticles they have been explored as diagnostics in point-of-care devices used to detect viral epitopes in HIV[72].

The genetic circuit is based on a split intein and σ factor to which the peptide library is fused. This could be used to screen/select

**Fig. 4 AMK-1057 is a cyclic peptide that binds human-derived Spike RBD in vitro. a** Schematic of the construct used to express AMK-1057. Leader and core amino acids are shown and predicted modified core residues highlighted in light blue. TEV protease sites are annotated as TEVp sites. **b** The expression, modification, cleavage, and purification steps are shown. TEV protease cleavage removes the SUMO tag/leader peptide and HPLC purification produces the final product; it leaves a single G, shown as a darker amino acid in the core. **c** High-resolution MS of unmodified AMK-1057 (black trace) and singly modified AMK-1057 (orange trace). **d** High-resolution MS/MS of modified AMK-1057 and fragment mapping to the amino acid sequence. Numbered peaks correspond to fragment ions observed and represented as lettered amino acids next to the MS/MS spectrum. **e** Structural annotation of AMK-1057. Thioether modification is colored to highlight the macrocycle. See Methods for details. **f** Binding of purified peptides (modified AMK-1057, blue; unmodified AMK-1057, gray; ACE2α1, orange) to human cell line-derived Spike RBD$_{296-531}$. Three replicates performed on different days are shown. **g** Space-filling models of the Spike RBD (PDB ID: 6M17) colored to indicate annotated binding regions of ACE2 (pink)[66], as well as antibodies CR3022 (orange)[67] and B38 (blue)[68]. Images were generated using ChimeraX[82]. The model is rotated 180°. **h–j** Competition BLI experiments measuring binding of purified, modified AMK-1057 to human cell-derived Spike RBD pre-incubated with **h** soluble hACE2 ($\Delta\Delta$nm = 0.99 ± 0.02), **i** antibody B38 ($\Delta\Delta$nm = 0.81 ± 0.03), and **j** antibody CR3022 ($\Delta\Delta$nm = 0.71 ± 0.03). $\Delta$nm was measured at 600 s and normalized to no AMK-1057 control to account for competitor off-rates ($\Delta\Delta$nm). Dashed lines separate association reactions from dissociation reactions (Methods). For all experiments, three replicates were performed on different days and a representative trace is shown.

libraries of linear peptides or protein domains for binding to the target. Here, we decided to use modified peptides because they have advantages as pharmaceuticals, including more compact structures that are resistant to proteolysis and permeate the mammalian cell membrane[1]. Modified peptides have been approved for the treatment of cancer, inflammation, and infection and increasing numbers are entering all phases of clinical development for diverse indications[73–76]. For example, the FDA-approved HIV antiviral Enfuvirtide is a 36 amino acid (aa) linear peptide that binds to a transmembrane glycoprotein; however, it suffers from rapid proteolysis, thus requiring twice daily injections[77]. Crosslinking HIV-1 mimetic peptides makes them proteolytically-stable, acid-resistant, and orally bioavailable[78]. Further, RiPPs are a large class of natural products and the rules to diversify natural RiPPs or combine modifications to create synthetic ones are growing[26,40,79]. Unlike NRPS pathways, RiPPs are simple to fuse to the intein/σ factor domains and to create a large and diverse library of modified peptides. Combining molecular diversity creation with a selection circuit in the same cell enables massive libraries to be evaluated to populate pharmaceutical discovery pipelines with binders to a target-of-interest with minimal biochemical information.

## Methods
**Strains, media, and chemicals.** *E. coli* NEB 10-beta (C3019I, New England Bio-Labs, MA) was used for all routine cloning. *E. coli* Marionette-Clo[53] was used for all selection experiments. *E. coli* Marionette-X, a Marionette-compatible derivative of NEB Express (C2523I, New England BioLabs, MA), was used for large-scale peptide expression experiments. TB media (T0311, Teknova, CA) supplemented with 0.4% glycerol (BDH1172-4LP, VWR, OH, USA) was used for peptide expression and modification. 2xYT liquid media (B244020, BD, NJ) and 2xYT + 2% agar (B214010, BD, NJ) plates were used for routine cloning and strain maintenance. SOB liquid media (S0210, Teknova, CA) was used for making competent cells. SOC liquid media (B9020S, New England BioLabs, MA) was used for outgrowth. Unless noted otherwise, cells were induced with the following chemicals: cuminic acid (268402, Millipore Sigma, MO) added as 1000× stock (200 mM) in EtOH or DMSO; 3-oxohexanoyl-homoserine lactone (3OC6-AHL) (K3007, Millipore Sigma, MO) added as a 1000× stock (1 mM) in DMSO; anhydrotetracycline (aTc) (37919, Millipore Sigma, MO) added as a 1000× stock (100 µM) in DMSO; isopropyl β-D-1-thiogalactopyranoside (IPTG) (I2481C, Gold Biotechnology, MO) added as 1000× stock (1 M) in water. Cells were selected with the following antibiotics: carbenicillin (Carb, C-103-5, Gold Biotechnology, MO) added as 1000× stock (100 mg/mL in H$_2$O); kanamycin (Kan, K-120-10, Gold Biotechnology, MO) as 1000× stock (50 mg/mL in H$_2$O); spectinomycin (Spec, S-140-5, Gold Biotechnology, MO); and chloramphenicol (Cm, C-105-5, Gold Biotechnology, MO). Liquid chromatography was performed with Optima Acetonitrile (A996-4, Thermo Fisher Scientific, MA) and water (Milli-Q Advantage A10, Millipore Sigma, MO) supplemented with LC-MS Grade Formic Acid (85178, Thermo Fisher Scientific). The following solvents/chemicals were also used: Ethanol (V1001, Decon Labs, PA), Methanol (3016-16, Avantor, PA), Ammonium bicarbonate (A6141 Millipore Sigma, MO), dimethyl sulfoxide (DMSO) (32434, Alfa Aesar, MA), Imidazole (IX0005, Millipore Sigma, MO), sodium chloride (X190, VWR, OH), sodium phosphate monobasic monohydrate (20233, USB Corporation, OH), sodium phosphate dibasic anhydrous (204855000, NJ), guanidine hydrochloride (50950, Millipore Sigma, MO), tris (75825, Affymetrix, OH), TCEP (51805-45-9, Gold Biotechnology, MO), and EDTA (0.5 M stock, 15694,

USB Corporation, OH). DNA oligos and gBlocks were ordered from Integrated DNA Technologies (IDT) (CA).

**Plasmids and genes.** Plasmids pTHSS-1282, pDAA558, and pAMK-267 were constructed from the parental pTHSSe-44 backbone, which has a pSC101 origin variant (var 2) and ampicillin resistance[80]. Plasmids pTHSS-1282, pDAA-558, and pAMK-267 contain a flexible linker sequence (GSSRGGKGGPGGRGGVGGGGGIGG) between the peptide/GFP and Npu$^C$ regions. Plasmids pAMK-925, pTHSS-2132, pAMK-866, and pAMK-870 were constructed from the parental pTHSS-1458 backbone, which has a colE1* origin variant and a kanamycin resistance marker[80]. The plasmid pEG06_044 contains the PapB enzyme, a p15A origin of replication and spectinomycin resistance. The parental backbone pTHSS-2012, which has a p15a origin and spectinomycin resistance was used for additional cloning experiments[80]. The plasmids pAMK-926 and pTHSS-2137 that contain the P$_{Lux}$ promoter expressing Npu$^C$-σ$^C$ and PMI-Npu$^C$-σ$^C$, respectively, were constructed from pTHSS-2012. The plasmids pAMK-925 and pTHSS-2132 that contain the P$_{Tac}$ promoter expressing σ$^N$-Npu$^N$ and residues 17–124 of Mdm2 (Mdm2*)-σ$^N$-Npu$^N$, respectively, were constructed from pTHSS-1458. The plasmid pAMK-870 that contains the constitutive P$_{J23105}$ promoter expressing Mdm2*-σ$^N$-Npu$^N$ and the P$_{20\_992}$ promoter expressing CAT-GFP was constructed from pTHSS-1458. The plasmid pAMK-866 that contains the constitutive P$_{J23105}$ promoter expressing 328-533 of the SARS-CoV-2 Spike protein (RBD)-σ$^N$-Npu$^N$ and the P$_{20\_992}$ promoter expressing CAT-GFP was constructed from pTHSS-1458. The peptide cloning plasmid pAMK-267, constructed from pTHSSe-44, contains the P$_{Lux}$ promoter upstream of an RBS-His tag-SapI-GFP-SapI-Npu$^C$-σ$^C$ where the *gfp* gene can be replaced by a peptide gene through Type IIs assembly methods using the enzyme SapI (NEB). The RBS from pAMK-267 was chosen from a library of RBS variants upstream of a His tag-PMI-Npu$^C$-σ$^C$ that was tuned for co-expression with constructs containing the P$_{J123105}$ promoter expressing Mdm2*-σ$^N$-Npu$^N$. The N-terminal His tag in pAMK-267 was left in place to provide a constant 11 aa for consistent levels of expression between different peptide sequences. The plasmid pAMK-670 that contains the P$_{Lux}$ promoter expressing PMI-Npu$^C$-σ$^C$ was constructed from pAMK-267. The plasmid pAMK-857 that contains the P$_{Lux}$ promoter expressing N-terminal residues 19-56 of ACE2 (ACE2α1)-Npu$^C$-σ$^C$ was constructed from pAMK-267. The plasmid pAMK-917 that contains the P$_{Lux}$ promoter expressing the hit sequence fused to Npu$^C$-σ$^C$ was constructed from pAMK-267. Note that the pTHSSe-44 and pTHSS-1458 backbones have origin variants that alter their copy numbers, making them approximately equivalent to a p15a backbone. Genes encoding Npu intein, PMI, Mdm2*, ACE2α1, and RBD were synthesized as gBlocks. The ECF20_992 gene was sourced from a previous publication[48].

**Cytometry analysis.** Fluorescence characterization was performed on a BD LSR Fortessa flow cytometer with the HTS attachment and BD FACSDiva software version 8.0.3 (BD, NJ). Samples were prepared by diluting overnight cultures 1:400 by adding 0.5 µl of cell culture into 200 µl of PBS containing 1 mg/mL Kan. Fluorescence measurements were made using the (488 nm) laser and all data was derived from the FITC-A channel (PMT voltage of 400 V). At least 30,000 events were collected for each sample, without gating, and the Cytoflow Python package (v 1.1.1) was used for downstream analysis. When presented, the median value is used.

**Evaluation of the split-intein σ factor circuit.** Strains of *E. coli* Marionette Clo harboring a combination of plasmids pTHSS-1282, pTHSS-2132, and pTHSS-2137 or pTHSS-1282, pAMK-925, and pAMK-926 were used for assessing intein splicing with or without PMI-Mdm2* induced association, respectively. Strains were grown in 1 mL of LB media + antibiotics for 20 h in a deep well 96-well plate (1896–2000, USA Scientific, FL) at 30 °C, 900 rpm in an Infors HT Multitron Pro (Infors USA, MD). Cultures were then diluted 1:100 into fresh 1 mL of LB media + antibiotics and serial 1:10 dilutions of inducers (IPTG, $10^{-3}$–$10^3$ µM; 3OC6-AHL, $10^{-3}$–$10^3$ nM) for 20 h in a deep well 96-well plate at 30 °C, 900 rpm in the

Multitron Pro. 0.5 µl of saturated cell culture were then diluted into 200 µl of PBS containing 1 mg/mL Kan for cytometry analysis.

**PMI variant induction profiling.** PMI variants without any N-terminal sequence additions were appended to the linker-Npu$^C$-σ$^C$ and inserted into the pDAA558 backbone using standard cloning methods. These plasmids were then co-transformed into a Marionette Clo strain alongside either pAMK-878 (off-target, Spike Mdm2*) or pAMK-877 (on-target, Mdm2*) and then grown overnight in 1 mL LB media + Kan/Carb at 30 °C, 900 rpm in a deep well 96-well plate. The next day, cultures were diluted 1:100 into 1 mL TB media + Kan/Carb containing either 0 or 1 µM 3OC6-AHL. After overnight growth in the same growth conditions, the variants were assayed as described in Cytometry Analysis.

**Two-hybrid assay for RBD/Mdm2* association.** To assay for protein–protein mediated splicing the following plasmid combinations were transformed into *E. coli* Marionette Clo and fluorescence was measured via cytometry: pAMK-866/pAMK-670 (RBD/PMI); pAMK-866/pAMK-857 (RBD/ ACE2α1); pAMK-870/ pAMK-670 (Mdm2*/PMI); pAMK-870/pAMK-857 (Mdm2*/ACE2α1). Strains were grown in 1 mL of LB media + antibiotics for 20 h in a deep well 96-well plate at 30 °C, 900 rpm in a Multitron Pro. Cultures were then diluted 1:100 into fresh 1 mL of LB media + antibiotics + 1 µM 3OC6-AHL (full induction of peptide plasmid) for 20 h in a deep well 96-well plate at 30 °C, 900 rpm in the Multitron Pro. 0.5 µl of saturated cell culture were then diluted into 200 µl of PBS containing 1 mg/mL Kan for cytometry analysis.

**Library generation.** The Pap library was designed with diversity at the ends and middle of the peptide and included either glutamate or aspartate as a cyclization partner, for a final sequence design of XCXXX[D/E]XCXXX[D/E]X. Using the degenerate nucleotide sequences NNK for any amino acid and GAW for aspartate or glutamate, we generated a library of $10^{12}$ peptides encoded by $10^{14}$ unique codon sequences. The library of plasmids lbAMK-103, which contains the P$_{Lux}$ promoter expressing the Pap library-Npu$^C$-σ$^C$ was constructed from pAMK-267. The *pap* library was amplified from pEG03_283 using degenerate oligonucleotides oAMK-915/916 (IDT). Gel purification was used to isolate the 124 bp amplicon, which was then cloned into pAMK-267 using the type IIS restriction enzyme SapI (NEB).

Linear insert and plasmid were mixed at a 1:1 molar ratio (200 fmol each) along with 10 µl 10× DNA ligase buffer, 2 µl T4 DNA ligase (HC) (20 U/µl) (M1794, Promega, WI) and 4 µl SapI in 100 µl total volume. Reactions were cycled 25 times for 2 min at 37 °C and 5 min at 16 °C then incubated for 30 min at 50 °C, 30 min at 37 °C, and 10 min at 80 °C in a DNA Engine cycler (Bio-Rad, CA). An additional 2 µl SapI was then added, and the assembly was incubated for 1 h at 37 °C. Assemblies were then purified using Zymo Spin I columns (Zymo Research, CA). Library assemblies were initially transformed into electrocompetent *E. coli* NEB 10β (C3020K, NEB, MA). We observed $1.5 \times 107$ colony forming units (CFU)/mL for lbAMK-103. Total transformants were estimated by colony counting after $10^7$-fold dilution and plating of liquid outgrowths on selective media.

**Calculation of the modified fraction of the library.** The initial, unselected *papA* library was transformed and plated to resolve individual colonies. A set of 19 random colonies were picked and sequenced via colony PCR. Of the 19 sequenced colonies, 18 were properly assembled. These 18 library members were then assessed for post-translational modification via LC-MS. Plasmids were transformed into either *E. coli* NEB Express or *E. coli* Marionette-X using 30 µL of competent cells and 1 µL of each plasmid being transformed in a PCR strip tubes (1402-4700, USA Scientific, FL or 951020401, Eppendorf, NY). Transformations were incubated on ice (20–30 min), heat shocked (42 °C, 30 s), and incubated on ice again (5 min). Cells were then transferred to a deep well 96-well plate (1896–2000, USA Scientific, FL) with 120 µL of SOC media. After outgrowth (Multitron Pro, 1 h, 30 °C) in an Infors HT Multitron Pro (Infors USA, MD), 900 µL LB was added with appropriate antibiotics (at 1.1× for 1× final concentration) and incubated (Multitron Pro, 30 °C, 900 r.p.m.) until all wells reached saturation (12–30 h). Overnight cultures were diluted 1:100 into 1 ml TB media with Carb/Kan in deep well plates. After 3 h incubation (Multitron Pro, 30 °C, 900 r.p.m.), inducer was added (1 mM IPTG, 100 µl cumate), and cultures were incubated for 20 h (Multitron Pro, 30 °C, 900 r.p.m.). To purify the peptides, the 96-well plates were centrifuged (Legend XFR, 4,500 g, 4 °C, 20 min), pellets were resuspended in 600 µL lysis buffer (5 M guanidinium hydrochloride, 50 mM sodium phosphate, pH 8), and freeze-thawed (frozen at −80 °C; thawed in Multitron Pro at 37 °C, 900 r.p.m.) Cell lysates were centrifuged (Legend XFR, 4,500 g, 4 °C, 60 min) and peptides affinity purified using His MultiTrap TALON plates (29-0005-96, GE Life Sciences (now Cytiva), MA), following manufacturer instructions, using 1 × 600 µL water and 2 × 600 µL lysis buffer for column equilibration, 4 × 600 µL wash buffer (50 mM Tris, 50 mM NaCl, 5 mM Imidazole pH 8), and 1 × 200 µL elution buffer (50 mM Tris, 50 mM NaCl, 150 mM Imidazole pH 8). TCEP (2 mM) was added to the peptides, they were digested with TEV protease (0.1 mg/mL) at 4 °C overnight (~16) and the samples were analyzed using the QTOF (see below). The 14 modified library sequences were then aligned and WebLogos generated (https://weblogo.berkeley.edu/logo.cgi) with default parameters, except without small sample correction.

**Selection of Pap library lbAMK-103.** Assembled library of plasmids lbAMK-103 was transformed into an electrocompetent Marionette Clo strain harboring the PapB modifying enzyme plasmid, pEG06_044, and the selection plasmid, pAMK-866 (all non-assembly transformation steps were > $1 \times 10^8$ efficiency). A 1 mL of liquid outgrowth of library transformants was diluted 1:50 in TB media + Carb/Kan/Spec + 1 µM 3OC6-AHL and 100 µM cumate to induce peptide + modifying enzyme, and grown at 30 °C, 250 r.p.m. for 20 h. For the first round of selection, cultures were then diluted 1:100 in TB Carb/Kan/Spec + 1 µM 3OC6-AHL and 100 µM cumate + 300 µM Cm and grown at 30 °C, 250 r.p.m. for at least 20 h (until cultures were saturated). A 0.5 µL aliquot was taken for cytometry analysis and 2 mL of culture was also taken to harvest plasmid. A 5 µL sample of purified plasmid was stored for NGS analysis and the rest was digested with 1 µL SapI (NEB) for 1 h at 37 °C to remove the background pEG06_044/pAMK-866 plasmid. The selected lbAMK-103 plasmid was then re-transformed into the strain of electrocompetent *E. coli* Marionette Clo strain harboring the PapB modifying enzyme, pEG06_046, and the selection plasmid, pAMK-866. The selection process was repeated once more with a Cm concentration of 800 µM and then once more with a Cm concentration of 1200 µM.

**NGS analysis.** Library construction for NGS was performed using the protocol for KAPA Hyper Prep Kits with PCR Library Amplification/Illumina series (KK8504, Roche). First, miniprepped library plasmids were amplified with Q5 polymerase (#M0492L, New England BioLabs, MA) with the primers oAMK-946/947 (Pap library) and oAMK-997/998 (Tgn/Lyn library). A 150 bp band was isolated via gel extraction. Indexed adapters were ligated and reamplified with 10 cycles of PCR. Gel extraction was then used to isolate the resultant 260 bp PCR product. Sample concentrations were calculated using a BioAnalyzer on a High Sense DNA chip (5067-4626, Agilent). Samples were diluted to 2 nM, denatured, and further diluted to 10 pM, with 10% phiX spike in. Samples were run on a HiSeq 2500 using HiSeq v2 reagents for Paired End Clustering and a 200 cycle SBS kit (PE-402-4002 and FC-402-4021, Illumina) using the Illumina HCS software version 2.2.68. Forward and reverse reads were both 110 cycles, with an 8 cycle single index read. Base-calling and demultiplexing were performed using the bcl2fastq software (Illumina) with default settings. After basecalling and indexing, sequences were identified and aligned using the leader sequence and then binned by sequence.

**Validation of sequences from NGS.** Hit peptides from NGS were resynthesized as gBlocks (IDT). These gBlocks were used as template for PCR to introduce SapI restriction sites compatible for re-cloning into the pAMK-267 library backbone. Newly reconstructed library members were transformed into Marionette-Clo cells containing modifying enzyme and selection plasmids and were then plated on media containing Carb/Kan/Spec. Individual transformants were then cultured in TB media + Carb/Kan/Spec in a deep well 96-well plate (1896-2000, USA Scientific, FL) and incubated overnight (Multitron Pro, 30 °C, 900 rpm). These cultures were then subcultured 1:100 in TB media + Carb/Kan/Spec either fully induced (1 µM 3OC6-AHL, and 100 µM cumate) or uninduced and incubated for 20 hr (Multitron Pro, 30 °C, 900 rpm) before taking 0.5 µL for standard flow cytometry analysis.

**Analysis of Pap2c_1 mutants.** Truncations of Pap2c_1 were generated via insertion into pAMK-267 with standard cloning methods. These variants were then transformed into a Marionette Clo strain with the PapB modifying enzyme plasmid, pEG06_044, and the selection plasmid, pAMK-866, and then grown overnight in 1 mL LB media + Kan/Carb at 30 °C, 900 rpm in a deep well 96-well plate. The next day, cultures were diluted 1:100 into 1 mL TB media + Kan/Carb containing either 0 or 1 µM 3OC6-AHL. After overnight growth in the same growth conditions, the variants were assayed as described in Cytometry Analysis.

**Site-saturation mutagenesis of Pap2c_1.** A library containing all possible single site substitutions was created from the TEV-GIRAY-core parent construct (pAMK-1056). 15 PCR reactions were run to create NNK substitutions at each position except C8 and E12. These 15 PCRs were combined in equimolar amounts and assembled into the pAMK-267 backbone using SapI (see Library Generation). Colony counting indicated a library size of $1.3 \times 10^6$. This library was then mini-prepped and transformed into the Marionette Clo strain harboring the PapB modifying enzyme plasmid, pEG06_044, and the selection plasmid, pAMK-866. After overnight growth in LB media 30 °C, the library with modifying enzymes and selection plasmid was then induced by diluting 20 µL into 1 mL TB media + Carb/Kan/Spec and 3.2 nM 3OC6-AHL. Overnight growth was conducted in a deep well 96-well plate at 30 °C with 900 rpm shaking. After the culture reached saturation (20 hours), a selection was conducted by diluting 10 µL into 1 mL TB media + Carb/Kan/Spec, 3.2 nM 3OC6-AHL, and 250 µM Cm. The library was also diluted into media without any Cm to provide a "pre-selection" library. These cultures were grown until saturated (20 hours) before being miniprepped. The pre- and post-selection library plasmids were then amplified via PCR using the osDAA154 and osDAA155 primers (Supplementary Table 3) and 200 bp bands we gel isolated. These fragments were then submitted for paired-end reads on an Illumina MiSeq, utilizing V2 chemistry. Paired-end alignments were conducted with PANDAseq and only sequences without any mismatches between forward and reverse reads

were kept for downstream processing, resulting in >50 K reads per sample. Read fractions for each core sequence variant were calculated by dividing the number of reads for each DNA sequence by total number of reads in the sample. Enrichment values were then calculated from these by dividing the read fraction of each sequence in the post-selection library by its read fraction in the pre-selection library. Degenerate DNA sequences corresponding to the same amino acid sequence were averaged to get a single enrichment value for each core variant. These amino acid sequence enrichments were then normalized to the enrichment for the parent core sequence to derive "Enrichment relative to parent".

**Peptide purification**. Peptide hit gBlocks were cloned into the peptide expression plasmid, pEG03-119 using their flanking SapI restriction sites. The peptide and modifying enzyme plasmids were co-transformed into *E. coli* Marionette-X, streaked onto 2xYT agar with Carb/Spec and incubated at 30 °C overnight. Individual colonies were used to inoculate 20 mL of LB media in a 125 mL shake flask and incubated overnight at 30 °C and 250 rpm in an Innova44 (Eppendorf, NY). A 5 mL aliquot of overnight starter culture was diluted in 500 mL total volume TB media with Carb/Spec in Fernbach flasks and grown at 30 °C and 250 rpm until reaching $OD_{600}$ 0.8–1.0, at which point 1 mM IPTG and 200 μM cumate are added. Induced cultures were grown for a further 20 h at 30 °C and 250 rpm and then centrifuged (4,000 g, 4 °C, 10 min) in a Sorvall RC 6+ centrifuge (Thermo Fisher Scientific, MA). Pellets were resuspended in 30 mL lysis buffer (5 M guanidinium hydrochloride, 300 mM NaCl, 10 mM imidazole, 50 mM sodium phosphate, pH 7.5), and freeze-thawed twice (frozen in −80 °C freezer; thawed in innova44 incubator at 30 °C, 250 rpm). Cell lysates were centrifuged (Eppendorf 5424, 20,000 g, 18 °C, 45 min) in a Sorvall RC 6+ centrifuge (Thermo Fisher Scientific, MA) and the peptides affinity purified via gravity-flow using 3 mL resin-bed volume of Ni-NTA agarose resin (88223, Thermo Fisher Scientific, MA), following manufacturer instructions, using 2 resin-bed volumes water and lysis buffer for column equilibration, 4 resin-bed volumes of wash buffer (5 M guanidinium hydrochloride, 300 mM NaCl, 25 mM imidazole, 50 mM sodium phosphate, pH 7.5), 4 resin-bed volumes of elution buffer buffer (5 M guanidinium hydrochloride, 300 mM NaCl, 250 mM imidazole, 50 mM sodium phosphate, pH 7.5). Eluate from Ni-NTA purification was then subjected to solid-phase extraction (SPE) using Strata-XL 500 mg tubes (8B-S043-HCH, Phenomenex, CA). The solid phase was first conditioned with 4 bed volumes of methanol and then water. Eluate was then loaded, washed with 8 bed volumes of 10 mM $NH_4CO_3$, and eluted with 8 bed volumes of 1:1 acetonitrile:aqueous 10 mM $NH_4CO_3$. Solvent was removed via lyophilization at −80 °C for 24–48 h. To cleave the SUMO and leader peptide from the core, the extracted peptide was resuspended in 20 mL TE buffer and 100 μl 20 mg/mL TEV protease and incubated overnight at room temperature with slow orbital shaking. The cleaved peptides were then desalted using a Strata-X PRO 500 mg SPE tubes (8B-S536-HCH, Phenomenex, CA). The solid phase was first conditioned with 4 bed volumes of methanol and then water. Eluate was then loaded, washed with 8 bed volumes of 10 mM $NH_4CO_3$, and eluted with 8 bed volumes of 1:1 acetonitrile:aqueous 10 mM $NH_4CO_3$. Solvent was removed via lyophilization at -80 °C for 24–48 h. After solvent removal, a 5 mL aliquot of the mixture resuspended in 10:90 acetonitrile:water was injected into a Agilent Technologies 1260 Infinity system HPLC (HPLC) system (Agilent Technologies, Santa Clara, CA) and separated using a 150 mm × 10 cm Aeris PEPTIDE XB-C18 column (100 Å, 5 μm) at a flow rate of 2 mL/min. Separation was carried out with a gradient program, with 0.1% formic acid as solvent A and acetonitrile with 0.1% formic acid as solvent B. The % B was held at 25% for 3 minutes, then increased to 50% over the next 17 min. The eluent was passed through a diode array detector (DAD) and absorbance at 270 nm was recorded. Detected peaks were collected using an Agilent G1364B Fraction Collector and again solvent was removed via lyophilization at −80 °C for 24–48 h. Samples were resuspended in 1 mL of 1:1 acetonitrile:aqueous 10 mM $NH_4CO_3$ in pre-weighed 2 mL microcentrifuge tubes (Eppendorf) and solvent was removed via lyophilization at −80 °C for 24–48 h. Yields were measured by comparing mass of empty tubes to tubes containing lyophilized powder.

**Liquid chromatography/mass spectrometry**. All chromatography was performed using the mobile phases ACN (acetonitrile supplemented with 0.1% formic acid and 0.1% water) and water (supplemented with 0.1% formic acid). The "QTOF" is an Agilent 1260 Infinity II liquid chromatograph with binary pump configured in low-dwell volume mode and column oven set to 40 °C, coupled to an Agilent 6545 QTOF mass spectrometer equipped with an Agilent electrospray ionization (ESI) source. Nitrogen gas is building-supplied and ESI source parameters are 350 °C gas temperature, 12 L/min gas flow, 30 psig nebulizer pressure, 350 °C sheath gas temperature, 8 L/min sheath gas flow, 3000 V capillary voltage, 1000 V nozzle voltage, 135 V fragmentor voltage, 15 V skimmer voltage, 600 V Oct 1 RF Vpp; the mass spectrometer was run in MS mode with reference mass enabled and tuned in positive mode with standard mass range (3200 $m/z$) and 2 GHz extended dynamic range. QTOF analysis was performed with a Phenomenex Aeris PEPTIDE XB-C18 2.6 μm 50 mm × 2.1 mm column. The flow rate was set at 0.5 mL/min and 5 μl sample was injected. The gradient used was 20% ACN for 0.5 min, 20–55% ACN over 5.5 min, 55–90% ACN over 0.5 minutes, 90% ACN for 1.5 min, with 0.8 min re-equilibration. LC–MS data were analyzed using MassHunter version 10.0. Accurate mass predictions of peptides were generated using the online resource, ChemCalc[81].

**Bio-layer interferometry**. Assays were performed on an Octet Red (ForteBio) instrument at 30 °C with shaking at 1,000 rpm. Ni-NTA biosensors (18-5101, ForteBio, NY) were hydrated in 1× kinetics buffer (diluted from 10× buffer; 18-5032, ForteBio, NY) for 30 min before the measurement. Expi293F human cell-derived and purified SARS-CoV-2 RBD ($RBD_{296-531}$) was loaded at 10–20 μg/mL in 1× Kinetics Buffer for 300 s prior to baseline equilibration for 180 s in 1× kinetics buffer. Association reactions of the peptide to $RBD_{296-531}$ were carried out in 1× kinetics buffer at various concentrations in a two-fold dilution series from 80 mM to 1.25 mM was carried out for 900 s. Then dissociation reactions were observed for 900 s. Data were acquired using the Octet Data Acquisition software version 9.0 and response data were generated using ForteBio data analysis software version 9.0.

**Competition analysis using Bio-layer interferometry**. Competition of peptide AMK-1057 with hACE2, B38, and CR3022 for binding to recombinant SARS-CoV-2 RBD was evaluated using a ForteBio Octet Red instrument at 30 °C with shaking at 1,000 rpm. Ni-NTA biosensors (18-5101, ForteBio, NY) were hydrated in 1X kinetics buffer (diluted from 10X buffer; 18-5032, ForteBio NY) for 30 min before the measurement. Expi293F human cell-derived and purified SARS-CoV-2 RBD was loaded at 20 μg/mL in 1X Kinetics Buffer for 450 s prior to baseline equilibration for 300 s in 1× kinetics buffer. The RBD-loaded biosensor tips were exposed (300 s) to the hACE2 soluble receptor (300 nM), B38 (300 nM), or CR3022 IgG (300 nM, and then exposed (300 s) to either AMK-1057 peptide (10000 nM) or 1X kinetics buffer. The data was interstep corrected using the FortéBio Data Analysis Software. Additional binding by the secondary molecule indicates an unoccupied epitope (non-competitor), while no binding indicates epitope blocking (competitor).

**Reporting summary**. Further information on research design is available in the Nature Research Reporting Summary linked to this article.

## Data availability
Source data are provided with this paper. Genetic part sequences are available in the Supplementary Information. Plasmids are available from Addgene. Any other data are available from the corresponding author upon reasonable request.

## Code availability
Python scripts used for cytometry processing are released as open source software under the MIT license (GitHub repository: https://github.com/VoigtLab/constr-peptide-paper-figs).

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

## Acknowledgements

This research was funded by the US Defense Advanced Research Projects Agency (DARPA) program award HR0011-15-C-0084, a research award from Novartis Institute for BioMedical Research (Cambridge, USA), and the Banting Fellowships Program (A.M.K.). This material is also based upon work supported by the National Science Foundation Graduate Research Fellowship under Grant no. #1745302 (D.A.A). Expi293F human cell-derived and purified SARS-CoV-2 RBD was obtained as a gift from the King lab (University of Washington). hACE2, B38, and CR3022 were obtained as a gift from the Schmidt lab (Harvard University).

## Author contributions

A.M.K., E.G., D.A.A., T.H.S.S., and C.A.V. conceived the study and designed the experiments. T.H.S.S. constructed initial split intein selection system. Z.Z. performed BLI experiments. D.L.N. performed pap2c library assessment experiments. A.M.K., Z.Z., and T.L.W. HPLC purified peptides. K.P. conducted NGS experiments. A.M.K. and A.C.E. cloned constructs, A.M.K., E.G., and D.A.A. performed all other experiments. A.M.K., E.G., D.A.A., and D.B.G. analyzed the data. A.M.K., E.G., D.A.A., and C.A.V. wrote the manuscript.

## Competing interests

The authors declare no competing interests.
