## [Peer Review File · Nature Communications]

Reviewers' Comments:

Reviewer #1:

Remarks to the Author:

In this manuscript the authors describe the development of a method for rapidly designing large libraries of modified peptides (using an E. coli host) that bind specifically to target proteins of interest. As a proof of principle, the authors then use their approach to identify a peptide capable of binding the SARS-CoV-2 Spike protein receptor binding domain with high nM affinity.

1. There are some issues with the referencing of the panels in the middle of Figure 1 (and the accompanying text explanations of the presented data) that make it extremely hard to follow what the authors are saying. Specifically:

a. The second paragraph of Page 5 references Figure 1d. I believe this should be referencing Figure 1e.

b. The legend of Figure 1e shows the equation used to calculate the fold change as $(Mdm2^* - PMI) / (\text{no bait} - \text{no peptide})$. This nomenclature makes it look like they are subtracting values, which is confusing (i.e. if they are subtracting something it isn't clear what). My interpretation is that they have just divided the relative fluorescence signals at the equivalent points in the binding vs no-binding conditions, in which case they should express the fold-change equation more clearly, for example using superscript – i.e. $RFUMdm2 + PMI / RFUNoBait + NoPrey$.

c. The authors state that the greatest fold difference observed is when σ_N -NpuN is fully induced and NpuC- σ_C is uninduced. However, Figure 1e seems to show the opposite (based on the labelling of the axes). Which is it, and how does this relate to the decision to use a constitutive promoter?

d. The discussion of the inducible range of the sensor when swapping out residues appears to relate to the data presented in Figure 1f. However, this is not referenced from the text at this point. Instead, Figure 1f is referenced when they mention that 'the peptide needs to be able to be modified by RiPP enzymes in the context of....', which does not seem to relate at all; I believe they mean to reference Figure 1g at this point.

e. Also, the authors mention that they could not reproduce ACE2 peptide binding to RBD as was originally reported; do they have a possible explanation for this?

f. When the authors actually do reference Figure 1g in the text, I'm not sure what they are actually meaning to reference; it seems to be a reference to a preliminary experiment, but 1g is simply a schematic. Do they mean to reference Figure 1 h,i and j?

2. Figure 2a needs to be more clearly defined. A small table identifying the function of each gene product would be helpful.

3. There are a few grammatical oddities throughout the paper. I recommend a careful review of language use.

4. The graphics/structure shown in Figure 1a and 1b don't seem to be necessary (they really don't add any immediately useful information to the paper). The text statements about the association of RBD with ACE2 and the related references seem more than sufficient.

5. That the 'Npu' shown in Figure 1c is the split intein being used should be explicitly defined in the Figure legend. Saying it is defined in the text isn't enough, particularly since it isn't explained at the point in the text where the panel is first referenced.

6. Label the N- and C- termini in Figure 1c.

7. The first paragraph on Page 5 refers should 3O6-AHL – this should be 3OC6-AHL.

8. Figure 2b legend should say that binding results in production of sfGFP-CAT (not just CAT).

9. In the discussion (and in general), the authors need to be more careful/clear with their wording. For example, the statement in the discussion that 'the binding target does not need to be known or expressed in a heterologous host' is confusing and can be misconstrued (i.e. it is not immediately apparent they mean the binding partner of the protein they are identifying peptide binders for; it could be misinterpreted as the protein they are targeting with peptide, which makes no sense).

Reviewer #2:

Remarks to the Author:

In this study, the authors present a new selection system to identify enzymatically cyclized peptides that bind to a target protein. The system relies on a split intein, with the two components genetically fused to the target (in this case the PSIKE RBD from SARS2-cov2) and to a library of peptides that are intended to be enzymatically cyclized and then bind to the target. Successful binding then leads to the intein system forming a sigma factor that reports on the binding. The authors use the system to identify a cyclized peptide that binds to the SPIKE RBD. In its present form, the paper has some interesting novel aspects but also leaves several key questions unanswered.

The authors point out the advantages of their system without highlighting disadvantages, one of which is rather critical. While selection systems that report on disruption of protein-protein interactions (PPI) are based on the intended activity (disrupt the PPI), selection systems such as the one used here are based on binding the target without assurance that this binding actually disrupts the PPI. The authors show their once-cyclized peptide binds to the SPIKE RBD, but they do not show that it disrupts ACE-SPIKE-RBD binding. It is certainly possible that their identified peptide binds SPIKE at a site entirely different from the ACE2 binding site, and hence that the selected peptide does not have the desired PPI disruption activity. For publication in a high impact journal, I feel that question should be answered with data (and this disadvantage of the current selection system needs to be discussed). While I agree with the authors that this study is a discovery exercise and hence that antiviral therapeutic data should not be a requirement, if the peptide binds the SPIKE protein at a site that would not prevent ACE2 binding, the discovery is not a starting point for follow-up studies such as optimization as the authors suggest.

The split intein system used to generate the selectable marker was reported previously by the authors, and hence the impact of the current study is in the formation of the library of cyclized peptides and its selection for binding to SPIKE-RBD to a site where it would disrupt the PPI with ACE2. For the former, the authors chose a system not previously used for library generation. They show that ~35% of a relatively small number of peptides is enzymatically modified by their enzyme. This means 5 peptides out of 14. The authors call this a "large fraction". I think to most researchers this would be a small fraction and a small number of peptides tested. Also not specifically mentioned is whether these were bicyclic peptides as intended, given that the final selected peptide was only monocyclic. The authors should provide MS data for the reader on these tests which is currently absent. Also, the Methods shows that 18 of the 19 sequences were correct but then mention only 14 peptides. Why did the number go down from 18 to 14? If the library has the intended diversity, one would not expect to obtain duplicate sequences unless the library is highly skewed towards a few sequences. At present the authors do not provide the sequencing data, which should be summarized in supporting data for the reader to judge the diversity of the library. Fig 1i,j only shows the diversity for the 14 peptides.

The tandem MS of the final product shows b12 and y2 ions. How do the authors explain this for the structure shown? Could the crosslink be to the Valine instead of Glu? The abstract of ref 63 mentions these types of enzymes also make crosslinks to other amino acids. Based on the current data, I am not convinced AMK-1057 has the structure proposed. Also, is it possible that the product from the selection is different from the structure of AMK-1057? Comparing the library design in Fig 1H with what the authors used to obtain AMK-1057 (introducing SUMO and a TEV site that is not in the library), the two are quite different. Does the incorporation of the TEV site

change the response in the selection system (i.e. does the peptide the authors used to make AMK-1057 in liter scale give the expected response when used in the selection system?). It would also be useful for the reader if the authors provided a figure of the system used to make AMK-1057. For instance, I was not sure if the authors replaced amino acids to introduce the TEV site or if they added the TEV site. I think from the text they added the sequence but would that not change the modification by the enzyme compared to the selection sequence? Also, for figure 3f, were the BLI measurements performed in the presence of reductants? The precursor for AMK-1057 contains two Cys that can form a disulfide during selection.

Specific comments:

Some references are incomplete (e.g. 61, 78).

Discussion: "This work introduces a technique to capture modified peptides that bind to a single target protein. There are several advantages over a two-hybrid screen, including that the binding target does not have to be known (or be a protein)". These two sentences both use the word target but they mean different things, which make it confusing. In the first the target is a viral protein and in the second sentence the authors mean its human binding partner. The first target has to be known for the current strategy to be used. For the second target indeed, that is not needed. But in its current wording, it is confusing.

Reviewer #1:

1. *There are some issues with the referencing of the panels in the middle of Figure 1 (and the accompanying text explanations of the presented data) that make it extremely hard to follow what the authors are saying. Specifically:*

a. *The second paragraph of Page 5 references Figure 1d. believe this should be referencing Figure 1e.*

We have edited the paper accordingly. Please note that Figure 1 has updated lettering now.

b. *The legend of Figure 1e shows the equation used to calculate the fold change as (Mdm2*-PMI)/(no bait-no peptide). This nomenclature makes it look like they are subtracting values, which is confusing (i.e. if they are subtracting something it isn't clear what). My interpretation is that they have just divided the relative fluorescence signals at the equivalent points in the binding vs no-binding conditions, in which case they should express the fold-change equation more clearly, for example using superscript – i.e. $RFUMdm2+PMI/RFUNoBait+NoPrey$.*

We have edited the figure caption (now Figure 1c) accordingly, which now reads (Fluorescence^{Mdm2*+PMI}/Fluorescence^{no bait+no peptide}).

c. *The authors state that the greatest fold difference observed is when σN -NpuN is fully induced and NpuC- σC is uninduced. However, Figure 1e seems to show the opposite (based on the labelling of the axes). Which is it, and how does this relate to the decision to use a constitutive promoter?*

We have edited the text to match the figure correctly (page 5, paragraph 2). We have included additional data (Supplementary Figure 1) that demonstrates the utility of maintaining the σN -NpuN under control of a constitutive inducer and varying the level of expression of NpuC- σC to discriminate tighter binders.

d. *The discussion of the inducible range of the sensor when swapping out residues appears to relate to the data presented in Figure 1f. However, this is not referenced from the text at this point. Instead, Figure 1f is referenced when they mention that ‘the peptide needs to be able to modified by RiPP enzymes in the context of.....’, which does not seem to relate at all; I believe they mean to reference Figure 1g at this point.*

We have edited the manuscript to reference Figure 1d (page 5-6 paragraph) in the context of swapping bait/peptide.

The referenced statement (page 6 paragraph 2) now refers to Figure 1e.

e. *Also, the authors mention that they could not reproduce ACE2 peptide binding to RBD as was originally reported; do they have a possible explanation for this?*

As part of in-depth structure-activity relationship experiments, we have addressed this question in the text (page 8, paragraph 2) and Figure 3. The affinity of the ACE2a1 peptide:RBD interaction is too low to detect in the assay as tuned.

We were able to approximately reproduce the binding of that ACE2 peptide to the RBD *in vitro* (page 9, paragraph 3; Figure 4f; Supplementary Figure 4).

f. When the authors actually do reference Figure 1g in the text, I'm not sure what they are actually meaning to reference; it seems to be a reference to a preliminary experiment, but 1g is simply a schematic. Do they mean to reference Figure 1 h,i and j?

We have edited the paper accordingly. Figure 1f illustrates the library design (page 6, paragraph 3). The diversity of the library is described in the same paragraph and represented in Figure 1g (with data in Supplementary Table 3).

2. Figure 2a needs to be more clearly defined. A small table identifying the function of each gene product would be helpful.

We have edited the legend to be more consistent/informative with respect to Figure 2a

3. *There are a few grammatical oddities throughout the paper. I recommend a careful review of language use.*

We have edited the paper accordingly.

4. *The graphics/structure shown in Figure 1a and 1b don't seem to be necessary (they really don't add any immediately useful information to the paper). The text statements about the association of RBD with ACE2 and the related references seem more than sufficient.*

We have removed this as the focus of the paper has shifted.

5. *That the 'Npu' shown in Figure 1c is the split intein being used should be explicitly defined in the Figure legend. Saying it is defined in the text isn't enough, particularly since it isn't explained at the point in the text where the panel is first referenced.*

We have modified the figure caption to describe the split intein used and described splicing events leading to transcription. Note this is now Figure 1a.

6. *Label the N- and C- termini in Figure 1c.*

The labels have been added (Figure 1a).

7. The first paragraph on Page 5 refers should 3O6-AHL – this should be 3OC6-AHL.

This has been corrected here and throughout the manuscript.

8. Figure 2b legend should say that binding results in production of sfGFP-CAT (not just CAT).

This has been corrected.

9. In the discussion (and in general), the authors need to be more careful/clear with their wording. For example, the statement in the discussion that 'the binding target does not need to be known or expressed in a heterologous host' is confusing and can be misconstrued (i.e. it is

not immediately apparent they mean the binding partner of the protein they are identifying peptide binders for; it could be misinterpreted as the protein they are targeting with peptide, which makes no sense).

We have edited the discussion, as suggested.

Reviewer #2:

1. *The authors point out the advantages of their system without highlighting disadvantages, one of which is rather critical. While selection systems that report on disruption of protein-protein interactions (PPI) are based on the intended activity (disrupt the PPI), selection systems such as the one used here are based on binding the target without assurance that this binding actually disrupts the PPI. The authors show their once-cyclized peptide binds to the SPIKE RBD, but they do not show that it disrupts ACE-SPIKE-RBD binding. It is certainly possible that their identified peptide binds SPIKE at a site entirely different from the ACE2 binding site, and hence that the selected peptide does not have the desired PPI disruption activity. For publication in a high impact journal, I feel that question should be answered with data (and this disadvantage of the current selection system needs to be discussed). While I agree with the authors that this study is a discovery exercise and hence that antiviral therapeutic data should not be a requirement, if the peptide binds the SPIKE protein at a site that would not prevent ACE2 binding, the discovery is not a starting point for follow-up studies such as optimization as the authors suggest.*

Indeed, we find that the modified peptide that we find does not disrupt ACE2 binding (see second paragraph of the resubmission letter). Work performed to determine the epitope binding region is shown in Figure 4g-j. The paper has been substantially edited to focus on the genetic circuit we constructed to find peptidic hits to a single target and the selection strategy as a platform. The title and introduction have been re-written to better frame this platform and to take a step back from SARS-CoV-2. The discussion has been edited to better describe the advantages and disadvantages of our approach.

The split intein system used to generate the selectable marker was reported previously by the authors, and hence the impact of the current study is in the formation of the library of cyclized peptides and its selection for binding to SPIKE-RBD to a site where it would disrupt the PPI with ACE2. For the former, the authors chose a system not previously used for library generation. They show that ~35% of a relatively small number of peptides is enzymatically modified by their enzyme. This means 5 peptides out of 14. The authors call this a “large fraction”. I think to most researchers this would be a small fraction and a small number of peptides tested. Also not specifically mentioned is whether these were bicyclic peptides as intended, given that the final selected peptide was only monocyclic. The authors should provide MS data for the reader on these tests which is currently absent. Also, the Methods shows that 18 of the 19 sequences were correct but then mention only 14 peptides. Why did the number go down from 18 to 14? If the library has the intended diversity, one would not expect to obtain duplicate sequences unless the library is highly skewed towards a few sequences. At present the authors do not provide the sequencing data, which should be summarized in supporting data for the reader to judge the diversity of the library. Fig 1i,j only shows the diversity for the 14 peptides.

We have not described the split intein-based selection system previously. This is the first work where it appears.

We have repeated this experiment with more detailed workup (i.e. cleaved and measure high-resolution monoisotopic masses) to address this concern (Page 6, paragraph 3; Figure 1f,g; Supplementary Table 3). The library does have a large fraction of modified peptides as 14/16 (of 16/19 correctly assembled/non-stop codon – note that this decrease is not due to repeated sequences) demonstrated mass shifts consistent with formation of either one or two macrocycles.

The tandem MS of the final product shows b12 and y2 ions. How do the authors explain this for the structure shown? Could the crosslink be to the Valine instead of Glu? The abstract of ref 63 mentions these types of enzymes also make crosslinks to other amino acids. Based on the current data, I am not convinced AMK-1057 has the structure proposed.

The tandem MS of the final product does not show a b12 ion (nor did it in the manuscript submitted). The y2 ion was misannotated by an older R script we had obtained, thank you for the note. We have since developed an in-house python package for analysis that has eliminated this error.

To our knowledge, there is no evidence for PapB being able to modify anything other than Cys-Asp/Glu residues (we believe that the reviewer is referring to a different but related enzyme from that reference?). Ref. 58 does show that, for PapB, mutation of Asp residues to Asn/Ala residues in the peptide substrate, PapA, abrogates crosslinking.

This evidence supports the annotated structure.

Also, is it possible that the product from the selection is different from the structure of AMK-1057? Comparing the library design in Fig 1H with what the authors used to obtain AMK-1057 (introducing SUMO and a TEV site that is not in the library), the two are quite different. Does the incorporation of the TEV site change the response in the selection system (i.e. does the peptide the authors used to make AMK-1057 in liter scale give the expected response when used in the selection system?).

We have conducted extensive structure-activity relationship work to address this question. In the selection assay we have characterized the contribution of terminal amino acids from the leader and core peptide through truncations (Page 8; Figure 3; computational predictions in Supplementary Figure 3).

We demonstrate through this work that the TEV site does indeed change the response in the selection system and infer through characterization of the previously reported ACE2a1 peptide (both in the selection assay and *in vitro* through BLI) that the AMK-1057 structure purified and characterized *in vitro* is likely a weaker binder than the species we selected for. Please refer to page 8-9 as well as Figures 3 and 4.

It would also be useful for the reader if the authors provided a figure of the system used to make AMK-1057. For instance, I was not sure if the authors replaced amino acids to introduce the TEV site or if they added the TEV site. I think from the text they added the sequence but would that not change the modification by the enzyme compared to the selection sequence? Also, for figure 3f, were the BLI measurements performed in the presence of reductants? The precursor for AMK-1057 contains two Cys that can form a disulfide during selection.

We have edited the manuscript and included several schemes in figures 3 and 4 that illustrate the various fusion peptide architectures. The specific peptide sequence in question (for AMK-1057) is shown schematically in Figure 4a.

We did not add reductant to the assay for fear of denaturing the target Spike RBD.

Some references are incomplete (e.g. 61, 78).

This has been corrected.

Discussion: 'This work introduces a technique to capture modified peptides that bind to a single target protein. There are several advantages over a two-hybrid screen, including that the binding target does not have to be known (or be a protein)'. These two sentences both use the word target but they mean different things, which make it confusing. In the first the target is a viral protein and in the second sentence the authors mean its human binding partner. The first target has to be known for the current strategy to be used. For the second target indeed, that is not needed. But in its current wording, it is confusing.

The discussion has been rewritten and edited for clarity.

Reviewers' Comments:

Reviewer #1:

Remarks to the Author:

The authors have addressed all my concerns and I'm happy to accept this manuscript.

Reviewer #2:

Remarks to the Author:

In this revised manuscript, the authors addressed my previous concerns. As I had worried, they indeed found that their hit compound does not disrupt the protein-protein interaction they had intended to disrupt. To their credit, they demonstrate this now with data, and they propose alternative uses of cyclic peptide binders to therapeutic proteins where the site of binding is not of direct therapeutic value by suggesting uses as diagnostics or as PROTACS. I agree that these are alternative uses of their technology that are potentially valuable. They also provide now a balanced discussion of the pros and cons of their method.

The authors also addressed my other questions (some by experiments and some by pointing out that I misunderstood their set-up). They also improved their figures such that now the design is clear.

Although the study in the end did not achieve what the authors had intended, the methods described are in my opinion valuable and add to the growing toolbox of genetically encoded protein binding modalities. As such I recommend publication and I do not have any additional suggested revisions.

Reviewer #1:

The authors have addressed all my concerns and I'm happy to accept this manuscript.

We thank the reviewer for their previous edits and suggestions.

Reviewer #2:

In this revised manuscript, the authors addressed my previous concerns. As I had worried, they indeed found that their hit compound does not disrupt the protein-protein interaction they had intended to disrupt. To their credit, they demonstrate this now with data, and they propose alternative uses of cyclic peptide binders to therapeutic proteins where the site of binding is not of direct therapeutic value by suggesting uses as diagnostics or as PROTACS. I agree that these are alternative uses of their technology that are potentially valuable. They also provide now a balanced discussion of the pros and cons of their method.

The authors also addressed my other questions (some by experiments and some by pointing out that I misunderstood their set-up). They also improved their figures such that now the design is clear.

Although the study in the end did not achieve what the authors had intended, the methods described are in my opinion valuable and add to the growing toolbox of genetically encoded protein binding modalities. As such I recommend publication and I do not have any additional suggested revisions.

We thank the reviewer for their previous edits and suggestions.